# Degradation Profiling of Nardosinone at High Temperature and in Simulated Gastric and Intestinal Fluids

**DOI:** 10.3390/molecules28145382

**Published:** 2023-07-13

**Authors:** Bian-Xia Xue, Tian-Tian Yang, Ru-Shang He, Wen-Ke Gao, Jia-Xin Lai, Si-Xia Liu, Cong-Yan Duan, Shao-Xia Wang, Hui-Juan Yu, Wen-Zhi Yang, Li-Hua Zhang, Qi-Long Wang, Hong-Hua Wu

**Affiliations:** 1National Key Laboratory of Chinese Medicine Modernization, State Key Laboratory of Component-Based Chinese Medicine, Tianjin University of Traditional Chinese Medicine, 10 Poyanghu Road, West Area, Tuanbo New Town, Jinghai District, Tianjin 301617, China; xbx1525912710@163.com (B.-X.X.); 15591953691@163.com (T.-T.Y.); herushang2021@163.com (R.-S.H.); gwkke123@163.com (W.-K.G.); laijiaxin0715@163.com (J.-X.L.); l13781726625@163.com (S.-X.L.); duancy19990514@163.com (C.-Y.D.); wangshaoxia1@163.com (S.-X.W.); huijuanyu@tjutcm.edu.cn (H.-J.Y.); wzy0504@tjutcm.edu.cn (W.-Z.Y.); 2Tianjin Key Laboratory of Therapeutic Substance of Traditional Chinese Medicine, Tianjin University of Traditional Chinese Medicine, 10 Poyanghu Road, West Area, Tuanbo New Town, Jinghai District, Tianjin 301617, China

**Keywords:** nardosinone, *Nardostachys jatamansi* DC., simulated gastric fluid, simulated intestinal fluid, degradation, 2–deoxokanshone M

## Abstract

Nardosinone, a predominant bioactive product from *Nardostachys jatamansi* DC, is well-known for its promising therapeutic applications, such as being used as a drug on anti-inflammatory, antidepressant, cardioprotective, anti-neuroinflammatory, anti-arrhythmic, anti-periodontitis, etc. However, its stability under varying environmental conditions and its degradation products remain unclear. In this study, four main degradation products, including two previously undescribed compounds [2–deoxokanshone M (64.23%) and 2–deoxokanshone L (1.10%)] and two known compounds [desoxo-narchinol A (2.17%) and isonardosinone (3.44%)], were firstly afforded from the refluxed products of nardosinone in boiling water; their structures were identified using an analysis of the extensive NMR and X–ray diffraction data and the simulation and comparison of electronic circular dichroism spectra. Compared with nardosinone, 2–deoxokanshone M exhibited potent vasodilatory activity without any of the significant anti-neuroinflammatory activity that nardosinone contains. Secondly, UPLC–PDA and UHPLC–DAD/Q–TOF MS analyses on the degradation patterns of nardosinone revealed that nardosinone degraded more easily under high temperatures and in simulated gastric fluid compared with the simulated intestinal fluid. A plausible degradation pathway of nardosinone was finally proposed using nardosinonediol as the initial intermediate and involved multiple chemical reactions, including peroxy ring-opening, keto–enol tautomerization, oxidation, isopropyl cleavage, and pinacol rearrangement. Our findings may supply certain guidance and scientific evidence for the quality control and reasonable application of nardosinone-related products.

## 1. Introduction

*Nardostachys jatamansi* DC., which has been extensively used for centuries in Himalayan countries to treat neurological and cardiovascular diseases [1,2], presents bioactive constituents such as nardosinone, nardosinonediol, isonardosinone, desoxo-narchinol A, and aristolone, which play crucial roles in the plant’s therapeutic properties [2]. Among these constituents, nardosinone is integral for the authentication and quality evaluation of *N. jatamansi* [3,4,5], and it is stipulated to constitute at least 0.1% of the dried processed roots and rhizomes of the plant, according to the Chinese Pharmacopoeia [6]. Despite a plethora of pharmacological benefits of nardosinone, including anti-inflammatory [7], antidepressant [8], cardioprotective [9], anti-neuroinflammatory [10], anti-arrhythmic [11], and other potential pharmacological effects [12,13,14,15], its practical application as a clinical candidate drug is still limited. However, the roots and the rhizomes *N. jatamansi*, containing nardosinone as the major bioactive ingredient, frequently appear in several well-known Chinese patent prescriptions, such as the Wenxin Granule [16], Shensong Yangxin Capsule [17], and Songbuli Oral Solution [18].

Nardosinone (CAS No: 23720-80-1; molecular formula: C_15_H_22_O_3_; molecular weight: 250.1569) is a colorless crystal or a white amorphous powder with a chemical structure of nardosinane-type sesquiterpenoid that features a unique peroxide bridge between C–7 and C–11 [1]. Nardosinone can be derived from gansongone, an aristolane-type sesquiterpenoid that has been isolated from *Nardostachys* plants by exposure to air oxygen or through a liquid phase autoxidation (such as ^1^O_2_, ^3^O_2_, and H_2_O_2_) in the presence of different phenolic reagents (e.g., phenols, quercetin, and naphthol) under dark conditions [19]. The intrinsic five-membered peroxy ring in nardosinone is unstable and is prone to having a ring-opening reaction occur [20]. A key intermediate metabolic pathway may be undergone by degradation or structural modification of nardosinone. Liu et al. found that when dissolved in ethanolic solution, nardosinone is largely stable in an alkaline environment; in contrast, it rapidly degrades in acidic and high-temperature conditions. In a solid state, nardosinone is relatively more stable under the conditions of high temperature and high humidity compared with its unstable and readily biodegradable properties in the condition of strong light [21]. Furthermore, in a boiling methanol solution [20], nardosinone was shown to degrade into C_12_–norsesquiterpenoids (e.g., desoxo-narchinol A) or other nardosinane-type C_15_–sesquiterpenoids (e.g., nardosinonediol, isonardosinone), which have also been isolated from the same plant of *N. jatamansi*.

Given the above-mentioned unstable property of nardosinone, it is questionable whether nardosinone exerts the therapeutic effects in a prototype form or not for the medicinal herbs and patent prescriptions that have nardosinone as the main component. Thus, the chemical unstable property is suggested to require adequate consideration in future in vivo investigations on the pharmacokinetic characteristics of nardosinone when orally administered. In light of this, two pharmacokinetic studies of nardosinone were reported using intravenous injection [22] and oral ingestion [23] as the routes of administration. However, to date, the metabolites of nardosinone have not been characterized and investigated in depth.

In light of this, our study aimed to isolate and identify the main degradation products of nardosinone after being refluxed in boiling water by combining several modern chromatographic separations with extensive spectroscopic analyses. Moreover, we attempted to profile the plausible degradation pattern of nardosinone by means of ultra-high performance liquid chromatography-photo-diode array (UPLC–PDA) and ultra-high performance liquid chromatography-diode array detector-quadrupole time-of-flight mass spectrometry (UHPLC–DAD/Q–TOF MS). The overall goal was to provide guidance and scientific evidence for future studies on the quality control and reasonable application of nardosinone-related products. The workflow of the entire study is presented in Figure 1.

## 2. Results and Discussion

### 2.1. Isolation and Structural Elucidation of the Main Degradation Products of Nardosinone

Column chromatography (ODS, silica gel, and Sephadex LH-20), preparative thin-layer chromatography (TLC), and recrystallization methods were combined for the chemical investigation of the refluxed products of nardosinone in boiling water (Figure 2); this led to the isolation and identification of two novel compounds (2–deoxokanshone M (**1**) and 2–deoxokanshone L (**3**)) and two known compounds [desoxo-narchinol A (**2**) and isonardosinone (**4**)] with yields of 64.23%, 1.10%, 2.17%, and 3.44%, respectively. 

2–Deoxokanshone M (**1**) was afforded as a colorless crystal or white amorphous powder and could be easily dissolved in methanol instead of dichloromethane. Its molecular formula of C_12_H_16_O_2_ was established based on the analysis of its HRESI–MS (high-resolution electrospray ionization mass spectrometry) ([M+H]^+^ *m*/*z* 193.1205, calcd 193.1223; [M-H] ^−^*m*/*z* 191.1066, calcd 191.1078) and NMR (nuclear magnetic resonance) data (Table 1). The IR (infrared spectroscopy) spectrum (Figure 3a) showed the presence of hydroxyl (*ν*_max_: 3331 cm^−1^) and conjugated double bonds (*ν*_max_: 1584 cm^−1^ and 1546 cm^−1^) in compound **1**. The ^1^H and ^13^C NMR and HSQC (heteronuclear single-quantum correlation) data (Figure 3 and Table 1) of compound **1** revealed the existence of two methyls [*δ*_H_ 0.92 (3H, s) and 0.87 (3H, d, *J* = 6.8 Hz); *δ*_C_ 19.4 and 15.4], three methylenes [*δ*_H_ 2.28 (2H, m), 2.20 (2H, m), and 1.46 (2H, m); *δ*_C_ 49.2, 25.6, and 25.3], two olefinic methines [*δ*_H_ 6.49 (1H, t, *J* = 4.0 Hz) and 5.23 (1H, s); *δ*_C_ 130.6 and 102.0], one aliphatic methine [*δ*_H_ 1.62 (1H, m) and *δ*_C_ 38.5], and four quaternary carbons (*δ*_C_ 196.3, 167.7, 136.1, and 37.4). The NMR data indicated that compound **1** was a C_12_ norsesquiterpenoid. Comprehensive analysis of ^1^H–^1^H COSY (^1^H–^1^H correlation spectroscopy), HSQC, and HMBC (heteronuclear multiple bonding correlation) spectra of compound **1** resulted in the full assignment of its ^1^H and ^13^C NMR data, as shown in Table 1. The ^1^H–^1^H COSY spectrum suggested the presence of one spin coupling system (H–1 to H–4 and H_3_–11), as shown in Figure 3f and Figure 4a. In the HMBC spectrum, the correlations between H_2_–2 and C–1, C–3, C–4, C–9, and C–10; between H–4 and C–3, C–5, C–6, C–11, and C–12; between H–6 and C–5, C–7, C–8, C–10, and C–12; and between H–8 and C–6, C–7, C–9, and C–10 established the chemical scaffold of 4–hydroxy-8,8a-dimethyl-6,7,8,8a-tetrahydronaphthalen-2(1*H*)-one ring for compound **1** (Figure 3e and Figure 4a). The absolute configuration of compound **1** was determined by a comparison of its experimental ECD (electronic circular dichroism) spectra with those calculated for 4*R*,5*S*–**1** and 4*S*,5*R*–**1** using the TDDFT (time-dependent density functional theory) method [24]. The experimental UV (ultraviolet absorption spectrum) (Figure 4b) and ECD (Figure 4c) curves of compound **1** matched well with the simulated ones for 4*R*,5*S*–**1**. Furthermore, the X–ray diffraction data (CuK*α*) analysis of a single crystal of compound **1** allowed for its unambiguous assignment of absolute configuration as 4*R*,5*S* (Figure 4d). Compound **1** represents a new C_12_ norsesquiterpenoid featuring an *α*, *β*-unsaturated enol group in its structure.

The other three degradation products were identified through comparisons of their NMR data with those reported for desoxo-narchinol A [25], 7–oxonardosinoperoxide [8], kanshone L [26], and isonardosinone [10], and their structures were determined as desoxo-narchinol A (**2**), 2–deoxokanshone L (**3**), and isonardosinone (**4**), respectively. Among them, 2–deoxokanshone L (**3**) came to be a 2–deoxidized derivative of kanshone L and a 7–deoxidized product of 7–oxonardosinoperoxide [8], and it represents a new nardosinone-type enol sesquiterpenoid that has not been previously isolated from *N. jatamansi*.

2–Deoxokanshone L (**3**) was isolated as a white amorphous powder with a molecular formula of C_12_H_22_O_3_ as established using an analysis of the HRESI–MS ([M+H]^+^ *m*/*z* 251.1647, calcd 251.1642) and NMR data (Table 2). The IR spectrum (Figure 5a) indicated the existence of hydroxyl (*ν*_max_: 3379 cm^−1^) and conjugated double bonds (*ν*_max_: 1571 cm^−1^ and 1609 cm^−1^) in compound **3**. The ^1^H and ^13^C NMR and HSQC data (Figure 5 and Table 2) of compound **3** revealed the existence of four methyls [*δ*_H_ 0.89 (3H, d, *J* = 7.8 Hz), 0.90 (3H, s), 0.98 (3H, s) 1.10 (3H, s); *δ*_C_ 16.7, 22.9, 32.9 and 25.2], two methylenes [*δ*_H_ 1.45 (2H, m) and 2.20 (2H, m); *δ*_C_ 25.8 and 25.5], two olefinic methines [*δ*_H_ 6.53 (1H, br s) and 5.16 (1H, s); *δ*_C_ 130.4 and 101.7], one aliphatic methine [*δ*_H_ 2.68 (1H, m) and *δ*_C_ 32.1], and five quaternary carbons (*δ*_C_ 199.0, 167.0, 136.8, 71.7 and 40.1), suggesting that compound **3** may be a C_15_ sesquiterpenoid. Comprehensive analysis of the ^1^H–^1^H COSY, HSQC, and HMBC data of compound **3** resulted in the full assignment of the ^1^H and ^13^C NMR data, as shown in Table 2. The ^1^H–^1^H COSY spectrum determined the presence of one spin coupling system (from H–1 to H–4 and H_3_–14), as shown in Figure 5f and Figure 6a. The HMBC correlations from H_2_–3 to C–1, C–2, C–4, C–5, and C–14; from H–4 to C–2, C–5, and C–15; from H–6 to C–5, C–8, C–10, C–11, C–13, and C–15; and from H–8 to C–6, C–9, and C–10 established the planar structure of compound **3** (Figure 5e and Figure 6a). The absolute configuration of compound **3** was assigned by a comparison of its experimental ECD data with those calculated for 4*R*,5*R,6R*–**3** and 4*S*,5*R,6S*–**3** using the TDDFT method [24]. As a result, the experimental UV (Figure 6b) and ECD (Figure 6c) curves of compound **3** were identical with the simulated ones for 4*R*,5*R,6R*–**3**. Hence, the absolute configuration of compound **3** was assigned as 4*R*, 5*R*, *6R*.

### 2.2. Evaluation of the Vasodilatory and Anti-Neuroinflammatory Activities of Nardosinone and the Main Degradation Products

Previous studies have revealed that nardosinone possesses significant anti-neuroinflammatory [10] and vasodilatory (IC_50_ 254.59 μM) effects [27]. In this study, these two activities of nardosinone and its primary degradation product (2–deoxokanshone M, **1**) were evaluated based on our previously reported methods [26,27]. As shown in Figure 7, nardosinone could significantly inhibit the inflammatory factor NO production of the LPS-induced neuroinflammation model in BV–2 microglia cells, which is consistent with previous results reported (IC_50_: 37.82–74.21 μM) [10] and compared with minocycline (IC_50_ = 23.69 ± 2.01 μM) [28]. At the same time, 2–deoxokanshone M (**1**) failed to exhibit any inhibitory effect (Figure 7a,b). Other main degradation products, isonardosinone and desoxo-narchinol A, have been reported to exhibit significant anti-neuroinflammatory activities with IC_50_ values of 37.82–74.21 μM [10] and 3.48 ± 0.47 μM [29], respectively. As presented in Figure 7c, 2–deoxokanshone M displayed a vasorelaxation activity that was more potent than nardosinone. Specifically, the vasodilation rate of nardosinone was 74.57% at a concentration of 400 μM, while that of 2–deoxokanshone M already reached 80.39% at a lower concentration of 75 μM.

Given these results, it appears that the degradation mentioned above is beneficial for nardosinone in the treatment of vasodilation-related diseases, exemplified by hypertension [27], and is unfavorable for nardosinone in the aid of neuroinflammation-associated neurodegenerative disorders such as Alzheimer’s disease, parkinsonian syndromes, and amyotrophic lateral sclerosis [30]. 

### 2.3. Degradation of Nardosinone in Different Conditions

As shown in Figure 8, the UPLC analysis result of the refluxed products of nardosinone in slightly boiling water for 0, 30, 60, and 120 min indicated that nardosinone (**5**) was rather unstable in hot water and degraded entirely after 2 h of refluxing in boiling water. The main degradation products were 2–deoxokanshone M (**1**), desoxo-narchinol A (**2**), 2–deoxokanshone L (**3**), and isonardosinone (**4**). Hence, compounds **1**–**5** were selected as the marker compounds (Figure 9c) for further UPLC profiling of the degradation of nardosinone under a high-temperature (in 50% aqueous methanol at 80 °C, HT) condition and in simulated gastric (SGF) and intestinal (SIF) fluids in vitro. In consideration of the UV absorption wavelengths of the five marker compounds (Figure 9a), a wavelength of 270 nm was chosen. Figure 9b displays the UPLC stacked chromatograms of the five standard solutions and different incubation products of nardosinone in HT, SGF, and SIF conditions for 60, 48, and 12 h, respectively. Moreover, as shown in Figure 10, the influence of incubation time on the incubation products of nardosinone under three different environments (HT, SGF, and SIF) was studied. Notably, the degradation profile of nardosinone incubated in 50% aqueous methanol (80 °C) for 108 h was consistent with that of nardosinone refluxed in boiling water for 2 h, but markedly different from those in SGF and SIF for 108 h. Desoxo-narchinol A (**2**) and 2–deoxokanshone L (**3**) were the common degradation products in the HT, SGF, and SIF conditions. Isonardosinone (**4**) could not be detected in the incubation products of nardosinone in SGF. At the same time, 2–deoxokanshone M (**1**) could be found in the degradation products of nardosinone incubated in SIF for 12 h, where it then gradually disappeared with the incubation time.

During the 108-h degradation process of nardosinone in HT, SGF, and SIF, the concentrations of nardosinone and the four main products changed along with the incubation time, as shown in Figure 11. Nardosinone degraded significantly faster in HT and SGF than in SIF (Figure 11a), and nardosinone degraded entirely after 108 h of incubation in HT and SGF instead of a 79.66% degradation in SIF. In the beginning of the HT condition (Figure 11b), nardosinone rapidly degraded at a rate of 3.74% per hour within 24 h, then slowly degraded at a rate of 0.27% per hour, finally disappearing within 60 h. 2–Deoxokanshone M continuously increased within 84 h and then tended to obviously decrease in the remaining time. 2–Deoxokanshone L and isonardosinone were constantly increased during the whole degradation process, while desoxo-narchinol A was increased at a very slow speed with extremely low concentrations. In SGF (Figure 11c), nardosinone degraded at a rate of 1.15% per hour within 84 h, then slowly at a rate of 0.14% per hour, finally reaching the full degradation within 108 h; Desoxo-narchinol A firstly increased and then decreased; 2–Deoxokanshone M continuously increased; 2–Deoxokanshone L increased rather slowly in this process, which led to the speculation that conversion into this compound may occur in the clue of sustained temperature. In SIF (Figure 11d), nardosinone rapidly degraded at a rate of 4.26% per hour within 12 h and then slowly at a rate of nearly 0.27% per hour; nardosinone would not degrade completely even after 108 h of incubation. Isonardosinone sharply increased within 5 h, and then its concentration remained largely unchanged in the remaining time; desoxo-narchinol A constantly increased in this process, while 2–deoxokanshone M also tended to firstly increase and later decrease until it was imperceptibly beyond the detection of UPLC-PDA analysis. 2–deoxokanshone L increased at a very slow similar rate, similarly to that of the SGF condition.

As mentioned above, the production of 2–deoxokanshone M increased and then decreased in 50% MeOH under high temperature, indicating that this product may still be unstable and tended to be further degraded or transformed into other derivatives. Therefore, the preliminary stability test of this compound was undertaken as follows: an accurately weighed amount of 20.0 mg of 2–deoxokanshone M was dissolved in 40 mL of 50% aqueous methanol and SGF–A and then incubated at 80 °C and 37 °C, respectively, for 48 h. Some incubation products with retention times of 25.90 min and 26.32 min were gradually afforded under high temperature in 50% MeOH, while no products were found in the SGF–A incubation (Appendix A). Moreover, when 2–deoxokanshone M was subjected to boiling water and refluxed for 2 h without methanol (Appendix A), there was no degradation of transformation undertaken instead. Compared with nardosinone (Figure 8), 2–deoxokanshone M showed significantly high stability in boiling water (Appendix A). Combining the findings mentioned above with the NMR data (Appendix A), it could be speculated that the instability of 2–deoxokanshone M in 50% MeOH at high temperature may be attributed to the keto–enol tautomerization and methylation of the enol group due to the presence of MeOH; however, 2–deoxokanshone M seemed to be inert and insensitive to temperature only. Thus, 2–deoxokanshone M (**1a**) was proposed to be simultaneously transformed into three other products (**1b**, **2a**, and **2b**), as deduced by the quadruple *δ*_C-8_ peaks in the ^13^C NMR spectrum (Figure 12) of 2–deoxokanshone M in CD_3_OD. And, two compounds (**2a**, and **2b**) furtherly afforded two methylation products (**3a** and **3b**) when incubated in 50% MeOH at a high temperature of 80 °C for 48 h. The concentration ratio of compounds **1**, **2**, and **3** was approximately 7:4:1 based on the integral values of *δ*_H-8_ in the ^1^H NMR spectrum of 2–deoxokanshone M in CD_3_OD before and after heating in hot aqueous methanol (Figure 13).

In brief, nardosinone was more unstable under conditions of HT and SGF than in SIF. Isonardosinone was not detected in the incubation products of nardosinone in SGF, and 2–deoxokanshone L was more likely to be generated by continuous heating and seemed to be inert to pH variations. 2–Deoxokanshone M was relatively stable only in boiling water and was prone to causing a methylation of the enol group in hot aqueous methanol.

Furthermore, the degradation of nardosinone was investigated under the conditions of SGF–A (without pepsin) and SGF–B (without HCl), as illustrated in Appendix A. As a result, nardosinone degraded faster in an acidic environment than in a condition of solely pepsin, suggesting that nardosinone was more susceptible to the pH change.

### 2.4. UHPLC-DAD/Q-TOF MS Analysis and Proposed Degradation Pathway of Nardosinone

The degradation products of nardosinone in HT, SGF, SIF, SGF–A, and SGF–B were further analyzed using a UHPLC-DAD/Q-TOF MS analysis in the positive mode, with the total ion current (TIC) chromatograms being presented in Figure 14 and the detailed MS information being listed in Appendix A. Five compounds were identified based on the MS fragments, as presented in Appendix A, and the proposed fragmentation pathway was speculated accordingly, as presented in Appendix A. It is worth mentioning that nardosinonediol was characterized by UHPLC-DAD/Q-TOF MS through an MS comparison with that of the standard compound. Nardosinonediol was found to be only present in the incubation products of nardosinone in SIF. 

All identified compounds belong to the nardosinane-type sesquiterpenoids; therefore, they share some common ionic fragments in the similar fragmentation pathways. Take nardosinone as an example (Figure 15): A quasi-molecular ion at *m*/*z* 251.1646 [M+H]^+^ has been detected, and it is predisposed to lose an H_2_O (*m*/*z* 18) due to the instability of the peroxy ring, resulting in a characteristic fragment ion at *m*/*z* 233.1569. The presence of the carbonyl group in the structure leads to a further loss of the CO group (*m*/*z* 28) to afford an ion at *m*/*z* 205.1588. More importantly, the inverse Diels–Alder (DA) reaction is readily available for the specialized structure of nardosinane-type sesquiterpenoid, and the ion at *m*/*z* 233.1569 could further produce a characteristic fragment ion at *m*/*z* 191.1429 in the clue of cleavage of C_3_H_6_ (*m*/*z* 42). In addition, losses of C_3_H_4_ (*m*/*z* 40) and CH_2_ (*m*/*z* 14) from the ion at *m*/*z* 233.1569 afforded the ions at *m*/*z* 219.1377 and *m*/*z* 193.1229, respectively. Similarly, a series of ionic fragments were further produced at *m*/*z* 177.0905, 163.0750, 149.0960, 135.0805, 123.0442, 109.1010, 95.0854, and 81.0698. Interestingly, keto–enol tautomerization may occur throughout the MS spectrometric cleavage process of a nardosinane-type sesquiterpenoid. Surprisingly, nardosinone and 2–deoxokanshone L could both generate a fragment of 2–deoxokanshone M in their cleavage processes.

Based on the above-mentioned findings, a plausible degradation mechanism has been proposed, as shown in Figure 16. Initially, the five-membered peroxy ring in nardosinone (**5**) [8] makes it prone to a ring-opening reaction, which directly converts it to nardosinonediol [8]. Nardosinonediol remains generally stable in a neutral medium and is readily oxygenated to 7–oxonardosinone [8] or dehydrated to form kanshone A [8] in acidic and/or high-temperature conditions. This might be why nardosinonediol cannot be detected in the degradation products of nardosinone in HT and SGF except for in SIF. Kanshone A could undergo an epoxidation reaction to form isonardosinone (**4**) [8]. The epoxy three-membered ring in isonardosinone (**4**) is highly unstable in an aqueous acidic solution; it is readily exposed to a ring-opening reaction followed by a pinacol rearrangement and transforms into 7–oxonardosinone. This might explain why isonardosinone (**4**) cannot be found in the incubation products of nardosinone in SGF and SGF–A. Kanshone A was subject to a dehydration reaction to provide desoxo-nardosinanone H [8] or a cleavage of C_3_H_6_ to generate desoxo-narchinol A (**2**) [8]. 7–Oxonardosinone was susceptible to a keto–enol interconversion to form 2–deoxokanshone L (**3**) with an unsaturated hydroxyl group, which was followed by a cleavage of C_3_H_6_O to afford 2–deoxokanshone M (**1**). It is noteworthy that all of the compounds mentioned above have been reported to be isolated from the same plant of *N. jatamansi* except for the novel compounds, namely 2–deoxokanshone M (**1**) and 2–deoxokanshone L (**3**).

## 3. Materials and Methods

### 3.1. General Experimental Procedures, Reagents, and Materials

Chromatographic eluting fractions were monitored using a TLC analysis using silica gel 60 GF254 plates (5 mm × 10 mm, 10 mm × 20 mm, and 20 mm × 20 mm, 20 μm, Merck, Darmstadt, Germany). The spots were visualized by heating the plates after spraying them with 10% H_2_SO_4_ in ethanol. The 1D/2D NMR spectra were recorded on a Bruker AV-III spectrometer (600 MHz, Bruker, Zurich, Switzerland). Optical rotations were measured using a Rudolph AUTOPOL V polarimeter (Rudolph Research Analytical, Hackettstown, NJ, USA). ECD data were collected using a JASCO-815 ECD spectrometer (JASCO Co., Ltd., Tokyo, Japan). The HR-ESIMS data were obtained on a UPLC-Q Exactive Orbitrap Mass system (Thermo Scientific, Waltham, MA, USA). UV spectra were scanned on an Agilent Cary 60 UV-Vis spectrophotometer (Agilent, Palo Alto, CA, USA), and IR spectra were obtained on a Varian 640 FT-IR spectrometer (Thermo Fisher Scientific, Waltham, MA, USA). The X-ray diffraction experiment was performed using a Rigaku Xtalab P200 diffractometer (Rigaku Co., Ltd., Tokyo, Japan). The melting point was supplied by an SGW*^®^* X-4 microscopic melting point apparatus (Shanghai PRECISION Scientific Instrument Co., Ltd., Shanghai, China). Mobile phases were filtered through 0.45 μm microporous membranes and ultrasonically degassed before analysis.

For the profiling of the degradation products, monobasic potassium phosphate, porcine pepsin, and trypsin were purchased from Shanghai Yuanye Bio-Technology Co., Ltd., Shanghai, China. HCl and NaOH were obtained from Tianjin Damao Chemical Reagent Co., Ltd., Tianjin, China. Watsons Water was provided by Guangzhou Watson’s Food & Beverage Co., Ltd., Guangzhou, China. Chromatographic-grade acetonitrile and formic acid were obtained from Thermo Fisher Scientific, Waltham, MA, USA. Other solvents that were used in this work were of analytical grade and were purchased from Concord Technology Co., Ltd., Tianjin, China. Nardosinone was prepared in our lab with a UPLC purity of above 98%, as detailed in Appendix A. The deuterated solvents for the NMR analysis, including chloroform-*d* and dimethyl sulfoxide-*d*_6_, were purchased from Cambrige Isotope Laboratories, Inc, Andover, MA, USA. For the assay of anti-neuroinflammatory activity, Dulbecco’s modified Eagle’s medium (DMEM), fetal bovine serum (FBS), trypsin, penicillin, streptomycin, and lipopolysaccharide (LPS) were purchased from Gibco BRL (Grand Island, NY, USA). Minocycline was obtained from Beijing Solarbio Science & Technology Co., Ltd., Beijing, China. Cell Counting Kit-8 (CCK-8) was supplied by Dojindo, Beijing, China. The Nitric Oxide Content Assay Kit was obtained from Beyotime Biotechnology Co., Ltd., Shanghai, China. For the evaluation of the vasodilatory activity, sodium chloride (NaCl), sodium bicarbonate (NaHCO_3_), and D-glucose were purchased from Damao Chemical Reagent Factory (Tianjin, China). Magnesium sulfate heptahydrate (MgSO_4_·7H_2_O) was obtained from Tianjin Guangfu Technology Development Co. (Tianjin, China). Potassium chloride (KCl) was available from Boote (Tianjin, China) Chemical Trading Co. (Tianjin, China). Potassium dihydrogen phosphate anhydrous (KH_2_PO_4_) and crystalline calcium chloride (CaCl_2_·2H_2_O) were acquired from Tianjin Bailens Biotechnology Co. (Tianjin, China). The dimethyl sulfoxide (DMSO), U46619 (9,11–methanoepoxy PGH2), acetylcholine (Ach), and sodium nitroprusside (SNP) used in the experiment were purchased from Sigma-Aldrich (St. Louis, MO, USA). Lastly, isoflurane was sourced from Rayward Life Technology Co., Ltd. (Shenzhen, China).

### 3.2. NMR Spectroscopy

The NMR spectra were recorded on a Bruker AV-III spectrometer equipped with 5 mm probes at 298 K. Chemical shifts were provided on the *δ* scale and referenced to TMS at 0.00 ppm for proton and carbon. The coupling constants (*J*) are in hertz. The pulse conditions are presented in Appendix A. 

### 3.3. Isolation and Identification of the Main Degradation Products of Nardosinone (**5**)

Nardosinone (**5**, 10.00 g) was added into 10 L of distilled water and then reflux heated, which maintained slight boiling for 3 h. The reaction condition was monitored using TLC and UPLC detection to ensure that nardosinone was entirely degraded. The final degradation products were dried under reduced pressure to obtain an 8.125 g degradation mixture. Then, the mixture was subjected to an ODS reversed-phase column eluted with gradient MeOH–H_2_O (30:70, 40:60, 50:50, 60:40, 70:30, 85:15, and 100:0 for 5, 5, 30, 13, 27, 4, and 10 column volumes, respectively, *v*/*v*) to obtain 9 fractions (Fr. 1–9). Fraction 4 from the 50% MeOH elution, *viz.*, 2–deoxokanshone M (**1**), was obtained (5.773 g) with a yield of 64.23%. Fraction 2 (1.252 g, one fraction obtained from the 40% MeOH elution) was then subjected to a further silica gel column and was eluted with a gradient solvent of petroleum ether/ethyl acetate (100:0, 5:1, 4:1, 3:1, 2:1, 1:1, 1:2, and 100:0 for 3, 6, 4, 4, 6, 5, 4, and 5 column volumes, respectively, *v*/*v*) to obtain 8 subfractions (Fr. 2.1–Fr.2.8). Furthermore, subfraction Fr. 2.2 (0.590 g) was recrystallized in methanol to afford isonardosinone (compound **4**, 344 mg, yield of 3.44%); subfraction Fr. 2.4 (0.460 g) was purified by preparative TLC (petroleum ether: ethyl acetate = 4:1, *v*/*v*) to obtain desoxo-narchinol A (compound **2**, 217 mg, *R*_f_ = 0.23, yield of 2.17%). Subfraction Fr. 2.8 (0.748 g) was further purified by Sephadex LH–20 gel column chromatography (dichloromethane: methanol = 1:1, *v*/*v*) and recrystallization (methanol) to provide 2–deoxokanshone L (compound **3**, 110 mg, yield of 1.10%). The purities of all isolated compounds were determined to be above 98% by the UPLC analysis.

*2–Deoxokanshone M* (**1**): Colorless crystals or amorphous powder; m.p.: 207 °C; ${\left[\alpha \right]}_{D}^{25}$ +26.0 (*c* 1.0, MeOH); UV *λ*_max_ (MeOH) (log *ε*): 287 (2.89) nm, 322 (2.89) nm; IR (KBr) *ν*_max_ cm^−1^: 3331, 2958, 2923, 2361, 2338, 1584, 1546, 1366, 1235, 1133; ECD (*c* 0.125, MeOH) *λ*_max_ (Δ*ε*): 210 (−4.33), 278.5 (+1.03), 328 (−1.05); ^1^H NMR (DMSO-*d*_6_, 600 MHz) and ^13^C NMR (DMSO-*d*_6_, 150 MHz): see Table 1; HRESIMS *m*/*z*: 193.1205 [M+H]^+^ (calcd for C_12_H_17_O_2_, 193.1223) and 191.1066 [M-H]^−^ (calcd for C_12_H_15_O_2_, 191.1078).

*2–Deoxokanshone L* (**3**): Write amorphous powder. ${\left[\alpha \right]}_{D}^{25}$ −9.23 (*c* 0.65, MeOH); UV *λ*_max_ (MeOH) (log *ε*): 200.2 (2.35) nm, 297.1 (2.55) nm; IR (KBr) *ν*_max_ cm^−1^: 3379, 2968, 2925, 2349, 2337, 1607, 1571, 1462, 1384, 1227; ECD (*c* 0.5, MeOH) *λ*_max_ (Δ*ε*): 209.8 (−1.1), 246.2 (−0.61), 291.6 (2.79), 328.4 (−2.97); ^1^H NMR (DMSO-*d*_6_, 600 MHz) and ^13^C NMR (DMSO-*d*_6_, 150 MHz): see Table 2; HRESIMS *m*/*z*: 251.1647 [M+H]^+^ (calcd for C_15_H_22_O_3_, 251.1642).

*X-ray crystallographic analysis of* (**1**): A suitable colorless crystal (0.16 × 0.13 × 0.1 mm^3^), afforded by slow evaporation of a mixture solution (methanol: water = 2:1) of (**1**), was mounted on a Rigaku Xtalab P200 diffractometer equipped with CuK*a* radiation (*λ* = 0.154184 nm). The intensity data were collected using CrysAlisPro 1.171.39.46 software at 294 K. Crystal data: C_12_H_16_O_2_, molecular formula = 192.12, cell parameters: a = 12.7239 (5) Å, b = 8.0557 (4) Å, c = 40.0828 (10) Å, *α* = 90°, *β* = 93.402 (3)°, *γ* = 90°, *V* = 4101.2 (3) Å^3^; monoclinic space group P2_1_, D_calc_ = 1.192 g/cm^3^, Z = 2, F (000) = 1596.0; absorption coefficient *μ* = 0.684 mm^−1^. A total of 113,981 reflections were collected (15350 unique, R_int_ = 0.0855, R_sigma_ = 0.0776) in the range of 6.628° ≤ 2*θ* ≤ 155.356°. The final indices [I > 2*σ* (I)] were R_1_ = 0.0614, wR_2_ = 0.1781; final R indexes [all data] R_1_ = 0.1095, wR_2_ = 0.2247; goodness-of-fit F^2^ = 1.017; flack parameter = −0.07(14). The structure was solved by direct methods using the Olex2 software package (https://www.olexsys.org/olex2/). The crystallographic data of 2–deoxokanshone M has been deposited at the Cambridge Crystallographic Data Center (Deposition number: CCDC 2215280).

*ECD Calculation of* (**1** and **3**)*:* As reported in our previous study [31], stochastic conformational searches were firstly performed using CONFLEX 8 software to afford the low-energy conformers. Geometry optimizations and the frequency pre-calculations of the low-energy conformers were finished at the B3LYP/6-311+g (2d, p) basis set level in MeOH. Using the optimized conformers of compound **1** and **3**, 100 excitation states at the B3LYP/6–311+g (2d, p) level were calculated using an IEFPCM solvent model in MeOH with a half bandwidth of 0.45 eV. The calculation results of different conformers were Boltzmann averaged to simulate the ECD spectra after UV correction, which were finally extracted using SpecDis 1.70.1 software.

### 3.4. Evaluation of the Vasodilatory Activity

Using the method previously described [27], the intact endothelium aorta was used to measure the vasorelaxation activity of the isolated compounds.

### 3.5. Evaluation of the Anti-Neuroinflammatory Activity

As reported in a previous report [31], a CCK-8 assay was used to evaluate the effects of the isolated degradation products on the cell viabilities of BV–2 microglial cells, and a Griess method [32] was applied to evaluate their activities on the NO production in LPS–simulated BV-2 cells, with minocycline being selected as the positive drug.

### 3.6. Quantitative UPLC–PDA Analysis

A Waters Acquity UPLC^®^ H class plus system (Waters Corporation, Milford, MA, USA), equipped with a column heater, sample manager, quaternary solvent manager, and photo-diode array (PDA) detector, was employed to undertake the chromatographic separation using an Acquity UPLC BEH C_18_ column (2.1 mm × 100 mm, 1.7 µm) at 35 °C. The mobile phase comprised acetonitrile (A) and 0.1% formic acid aqueous solution (B). The samples were detected at a wavelength of 270 nm with the flow rate set at 0.3 mL/min and an injection volume of 3 µL, and the gradient program was set as follows: 0–22 min, 18–26% (A); 22–30 min, 26–95% (A), respectively.

For quantitative analysis, linearity was evaluated by analyzing six different concentrations of the standard solutions (nardosinone, isonardosinone, desoxo-narchinol A, 2–deoxokanshone L, and 2–deoxokanshone M). Each linearity sample was injected in triplicate. The calibration curve was constructed as a linear regression analysis of the peak area (*Y*–axis) versus the corresponding concentration (*X*–axis). The limits of detection (LOD) and quantification (LOQ) of the tested compounds were calculated as concentrations where the signal–noise ratio (S/N) was about 3 and 10, respectively. The calibration plot was linear for 5 evaluation components, and a linear relationship with the *r*^2^ values being greater than 0.9995 was observed in all cases. The linear relationship, LOD, and LOQ values are detailed in Table 3.

### 3.7. UHPLC–DAD/Q–TOF MS Analysis

The analysis was performed on an Agilent 1260 Infinity II UHPLC system equipped with an Agilent 6550 iFunnel Q–TOF MS detector (Agilent Technologies, Palo Alto, CA, USA). The chromatographic condition of UHPLC was employed as described in Section 3.6. For the MS analysis, nitrogen was used as drying gas at a flow rate of 12 L/min. The gas temperature and sheath gas temperature were fixed at 200 °C and 350 °C, respectively. The nebulizer was 40 psi. The capillary voltage and nozzle voltage were set as 3.0 kV and 1.5 kV, respectively. The spectra were recorded in the *m*/*z* 50–1200 range for a full scan.

### 3.8. Preparation of Simulated Gastrointestinal Fluids

Simulated gastric fluid (SGF) and simulated intestinal fluid (SIF) in vitro conditions were prepared according to the Chinese Pharmacopoeia (2020 Edition, Part IV) [33]. For SGF, 10 g of pepsin was dissolved in 16.4 mL of HCl and a sufficient volume of water to form up to 1000 mL with a pH of approximately at 1.3. For SIF, 6.8 g of monobasic potassium phosphate was dissolved in 500 mL of water with a pH adjusted to 6.8 using 0.1 mol/L NaOH solution. An amount of 10 g of trypsin was dissolved in an appropriate amount of water, and the two solutions were mixed; the final solution volume formed up to 1000 mL with water. In addition, simulated gastric fluid A (SGF–A, without pepsin) and B (SGF–B, without HCl) were also prepared to investigate the specific factors (acidic or not) affecting the degradation of nardosinone compared with that in SGF.

### 3.9. Incubations of Nardosinone at High-Temperature Condition and in Simulated Gastrointestinal Fluids

An accurately weighed amount of 20.0 mg of nardosinone was dissolved in 40 mL of 50% aqueous methanol, and the solution was then incubated at 80 °C for 108 h. Similarly, each accurately weighed 20.0 mg amount of nardosinone was dissolved in 40 mL of SGF, SIF, SGF–A, and SGF–B fluids, respectively, and incubated at 37 °C for 108 h. It is worth noting that due to the poor water solubility of nardosinone, it does not dissolve in simulated gastrointestinal fluids at room temperature; instead, it is in a suspended state. Therefore, an equal volume of methanol was added to dissolve and configure the sample before further tests or analyses. Moreover, during the incubations, the concentrations of nardosinone, isonardosinone, desoxo-narchinol A, 2–deoxokanshone L, and 2–deoxokanshone M were calculated based on the regression equation at each point of time (0, 2, 4, 6, 8, 10, 12, 24, 48, 60, 72, 84, 96, and 108 h) using the above-established UPLC method. The stability study of nardosinone was repeated three times to guarantee the reliability of the results. In addition, trivalent 1.0 mg of nardosinone was dissolved and incubated in 2 mL of 50% aqueous methanol (80 °C), SGF (37 °C), and SIF (37 °C), respectively, until it completely degraded to afford the incubation end products before being diluted five times with methanol for the UHPLC-DAD/Q-TOF MS analysis.

### 3.10. Statistical Analysis

The data were presented as the mean ± standard errors of the mean (SEM). The statistical differences were analyzed using GraphPad Prism 8.0 software using a one-way ANOVA test. Values of *p* < 0.05 were considered statistically significant.

## 4. Conclusions

Four main degradation products of nardosinone in refluxing boiling water were isolated and identified as 2–deoxokanshone M, desoxo-narchinol A, 2–deoxokanshone L, and isonardosinone. Among them, 2–deoxokanshone M represents a new C_12_ norsesquiterpenoid featuring an *α*, *β*-unsaturated enol group in its structure, and it possesses significant vasodilatory activity without any anti-neuroinflammatory activity. Nardosinone is more susceptible to temperature and pH changes, and the degradation products slightly vary in high-temperature and simulated gastric and intestinal fluids conditions. Furthermore, the transformation pathway of 2–deoxokanshone M was reasonably deduced, and the MS fragmentation pattern of nardosinone and its plausible degradation pathway were proposed accordingly. Our study preliminarily sheds light on the stability of nardosinone. Future investigations are still needed for the analysis on the metabolites of nardosinone and the components that are absorbed in the blood. This study of the degradation products and patterns of nardosinone will facilitate the identification of the components of *N. jatamansi* extracts or fractions and the elucidation of dynamic mechanisms, and these findings may contribute to providing reasonable guidance for the preparation, production, and quality control of nardosinone, *N. jatamansi*, and the prepared patent Chinese medicines that contain *N. jatamansi*.

## Figures and Tables

**Figure 1 molecules-28-05382-f001:**
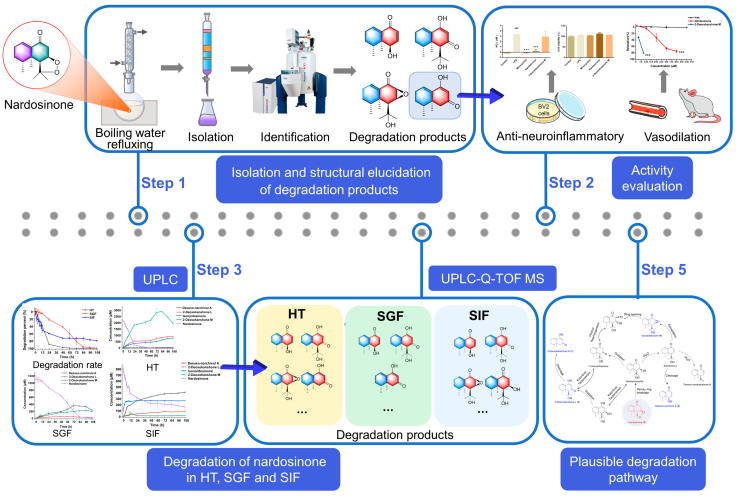
Workflow of the degradation profiling of nardosinone at high temperature and in simulated gastric and intestinal fluids. ^###^ *p* < 0.001 vs. control group, *** *p* < 0.001 vs. LPS group.

**Figure 2 molecules-28-05382-f002:**
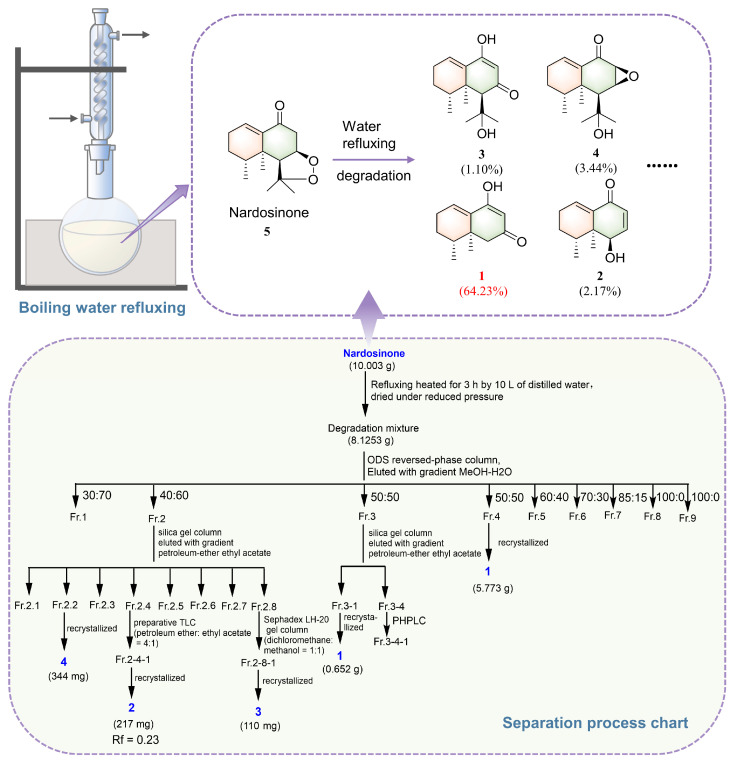
Separation process of the main degradation products of nardosinone in refluxing boiling water.

**Figure 3 molecules-28-05382-f003:**
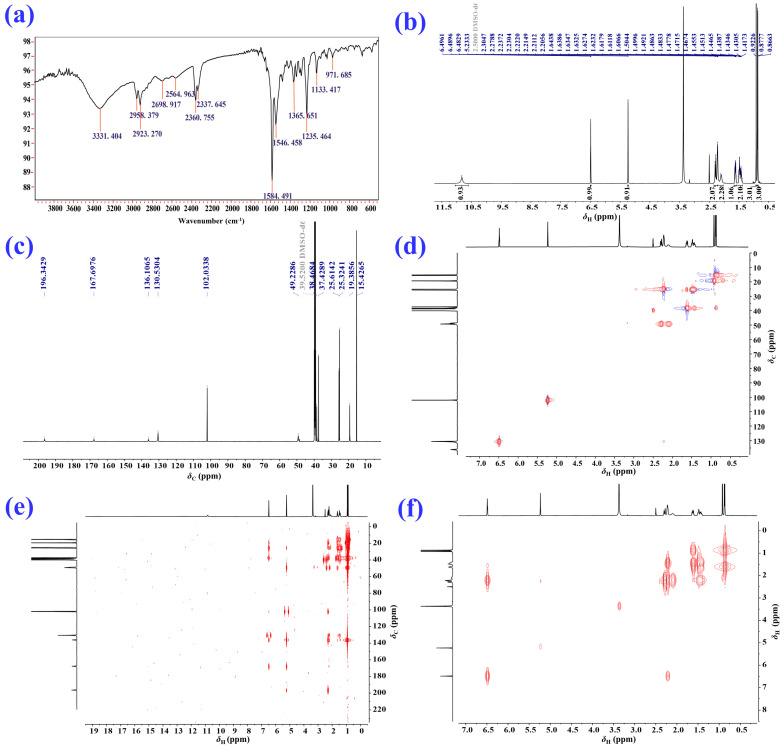
Spectrograms of 2–deoxokanshone M (**1**). (**a**) IR; (**b**) ^1^H NMR (600 MHz, DMSO–*d*_6_); (**c**) ^13^C NMR (150 MHz, DMSO–*d*_6_); (**d**) HSQC; (**e**) HMBC; (**f**) ^1^H–^1^H COSY.

**Figure 4 molecules-28-05382-f004:**
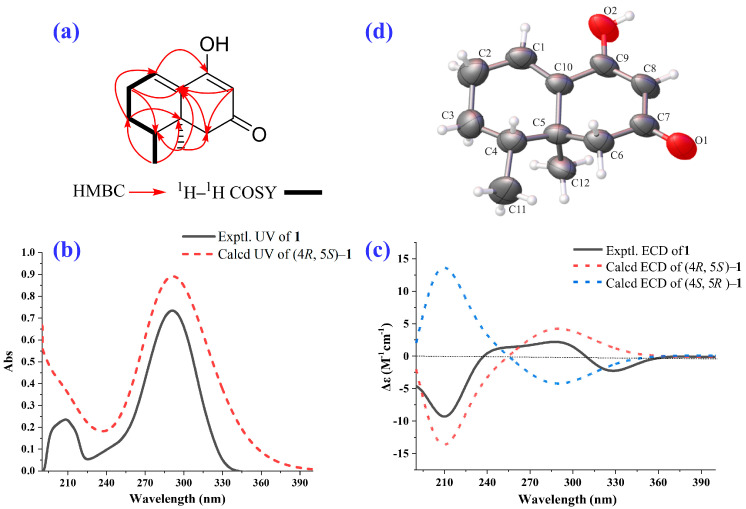
Structural identification of 2–deoxokanshone M (**1**). (**a**) Key HMBC and ^1^H–^1^H COSY correlations; calculated and experimental UV (**b**) and ECD (**c**) spectra; (**d**) ORTEP drawing established by the analysis of the single crystal X–ray diffraction data of compound **1**.

**Figure 5 molecules-28-05382-f005:**
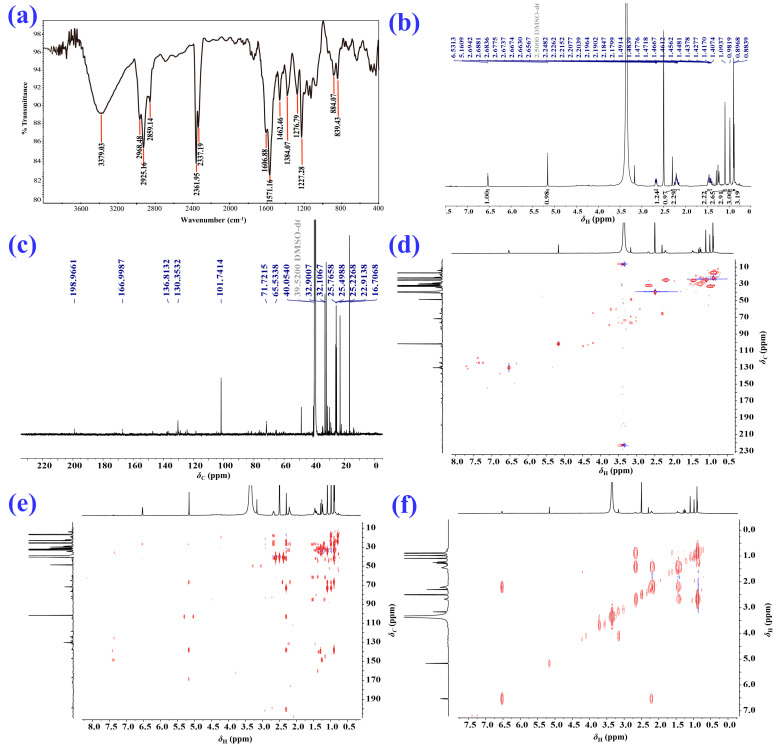
Spectrograms of 2–deoxokanshone L (**3**). (**a**) IR; (**b**) ^1^H NMR (600 MHz, DMSO–*d*_6_); (**c**) ^13^C NMR (150 MHz, DMSO–*d*_6_); (**d**) HSQC; (**e**) HMBC; (**f**) ^1^H–^1^H COSY.

**Figure 6 molecules-28-05382-f006:**
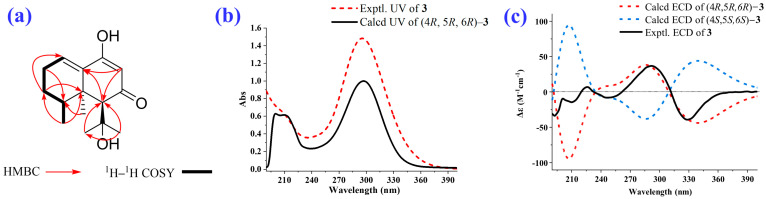
Structural identification of 2–deoxokanshone L (**3**). (**a**) Key HMBC and ^1^H–^1^H COSY correlations; calculated and experimental UV (**b**) and ECD (**c**) spectra.

**Figure 7 molecules-28-05382-f007:**
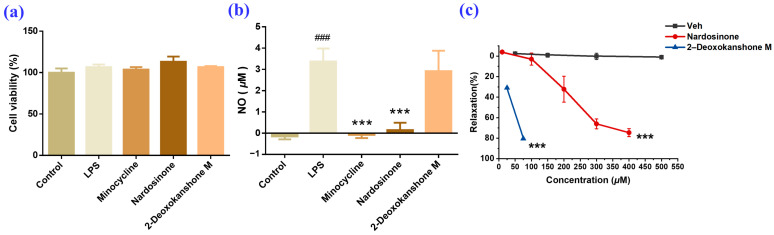
Anti-neuroinflammatory and vasodilatory activities of nardosinone and 2–deoxokanshone M. Effects on the cell viability (**a**) and production of NO in LPS–induced BV–2 cells (**b**) (^###^ *p* < 0.001 vs. control group, *** *p* < 0.001 vs. LPS group, n = 5); (**c**) dose-vasorelaxant activity (*** *p* < 0.001 vs. Veh., n = 3).

**Figure 8 molecules-28-05382-f008:**
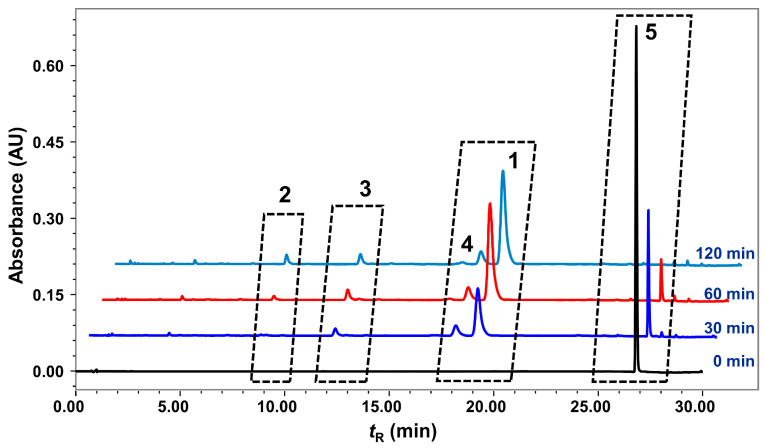
Stacked UPLC chromatograms of refluxed products of nardosinone in boiling water for 0, 30, 60, and 120 min.

**Figure 9 molecules-28-05382-f009:**
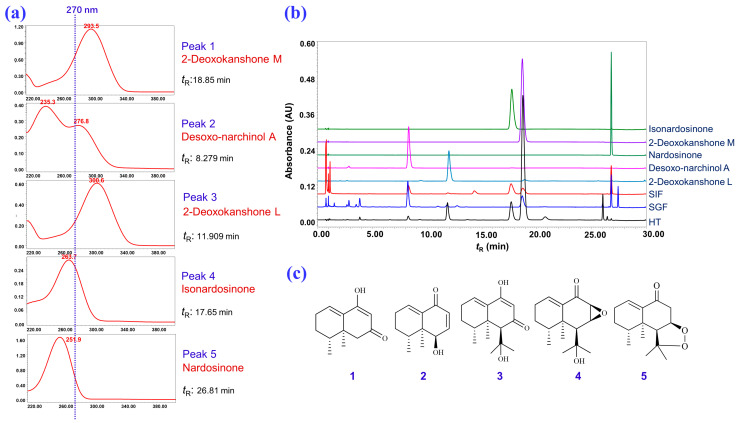
Attribution of the marker compounds. (**a**) UV absorption curves of the five quantitative marker substances (210–400 nm); (**b**) stacked UPLC chromatograms of the five quantitative marker substances and incubation products of nardosinone in HT, SGF, and SIF; (**c**) chemical structures of the five quantitative marker substances.

**Figure 10 molecules-28-05382-f010:**
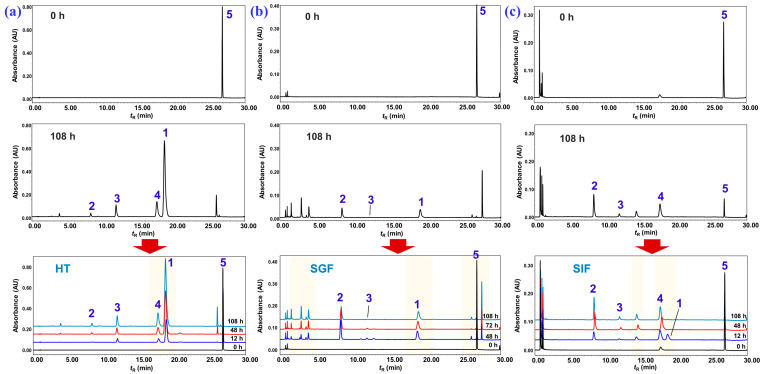
Representative UPLC chromatograms of the incubation products of nardosinone in HT, SGF, and SIF.

**Figure 11 molecules-28-05382-f011:**
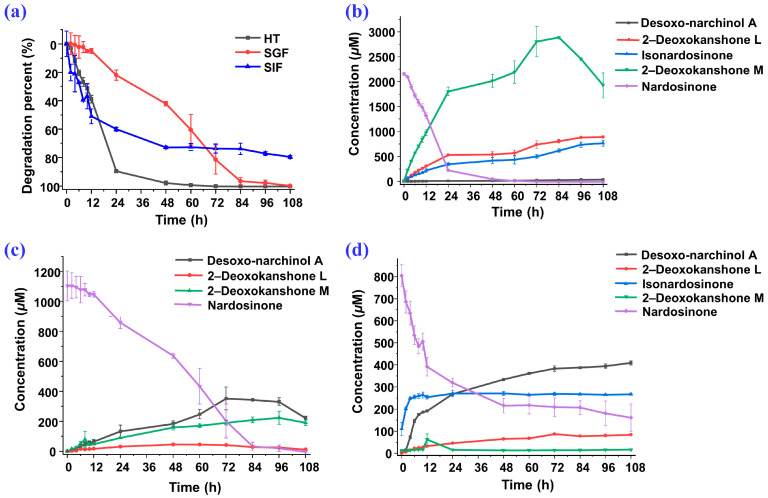
The degradation line charts of nardosinone and the degradation products under three different conditions for 108 h. (**a**) Degradation percents of nardosinone in three conditions; time–concentration curves of nardosinone and its main degradation products in HT (**b**), SGF (**c**), and SIF (**d**).

**Figure 12 molecules-28-05382-f012:**
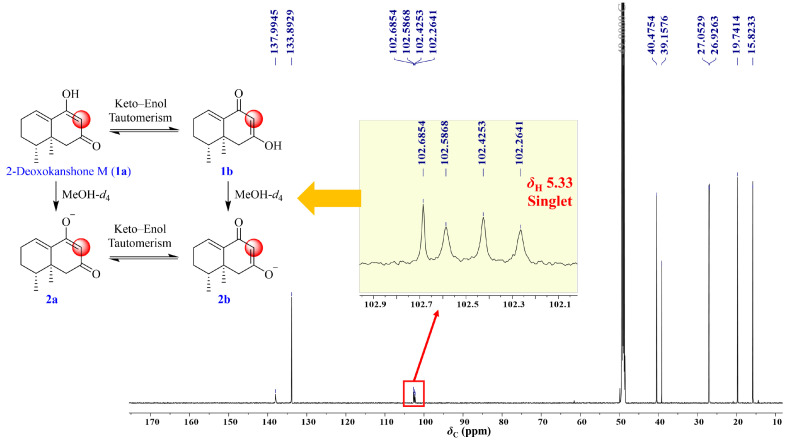
The plausible transformation pathway and ^13^C NMR spectrum of 2–deoxokanshone M in CD_3_OD.

**Figure 13 molecules-28-05382-f013:**
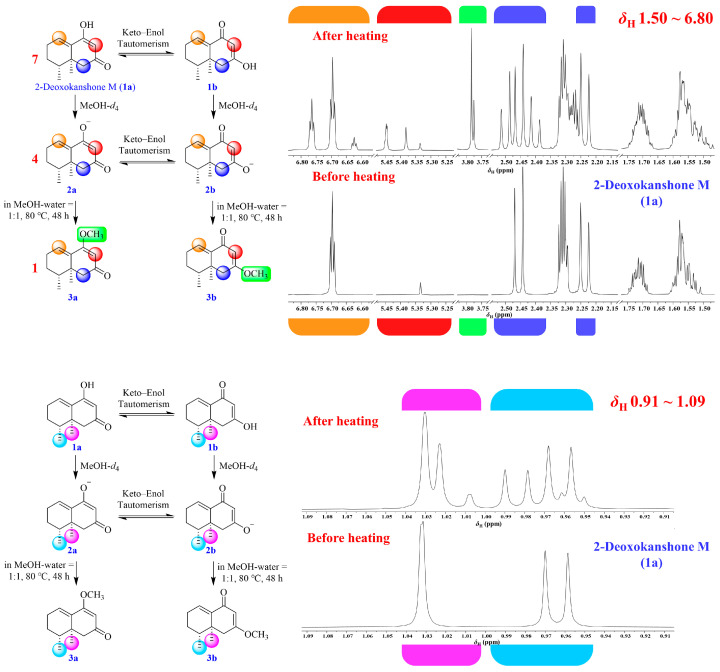
The plausible transformation pathway and ^1^H NMR spectra (in CD_3_OD) of 2–deoxokanshone M before and after heating in hot aqueous methanol.

**Figure 14 molecules-28-05382-f014:**
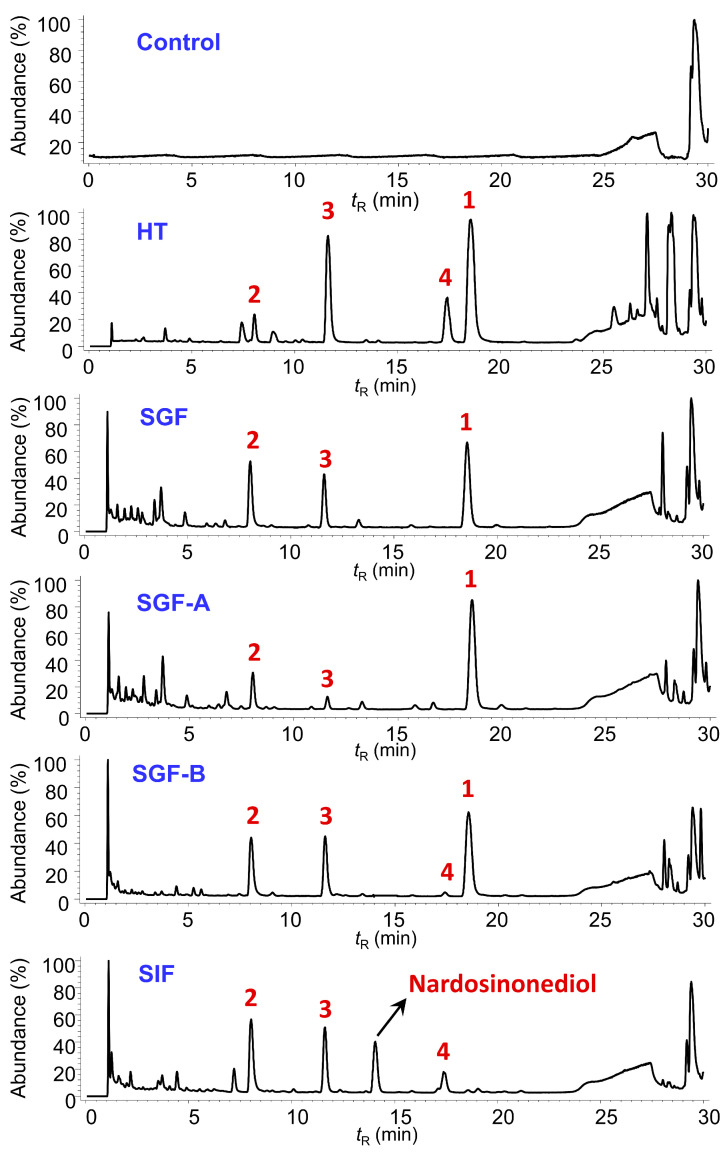
TIC chromatograms of the products of nardosinone incubated in different conditions by the (+)–UHPLC–DAD/Q–TOF MS analysis.

**Figure 15 molecules-28-05382-f015:**
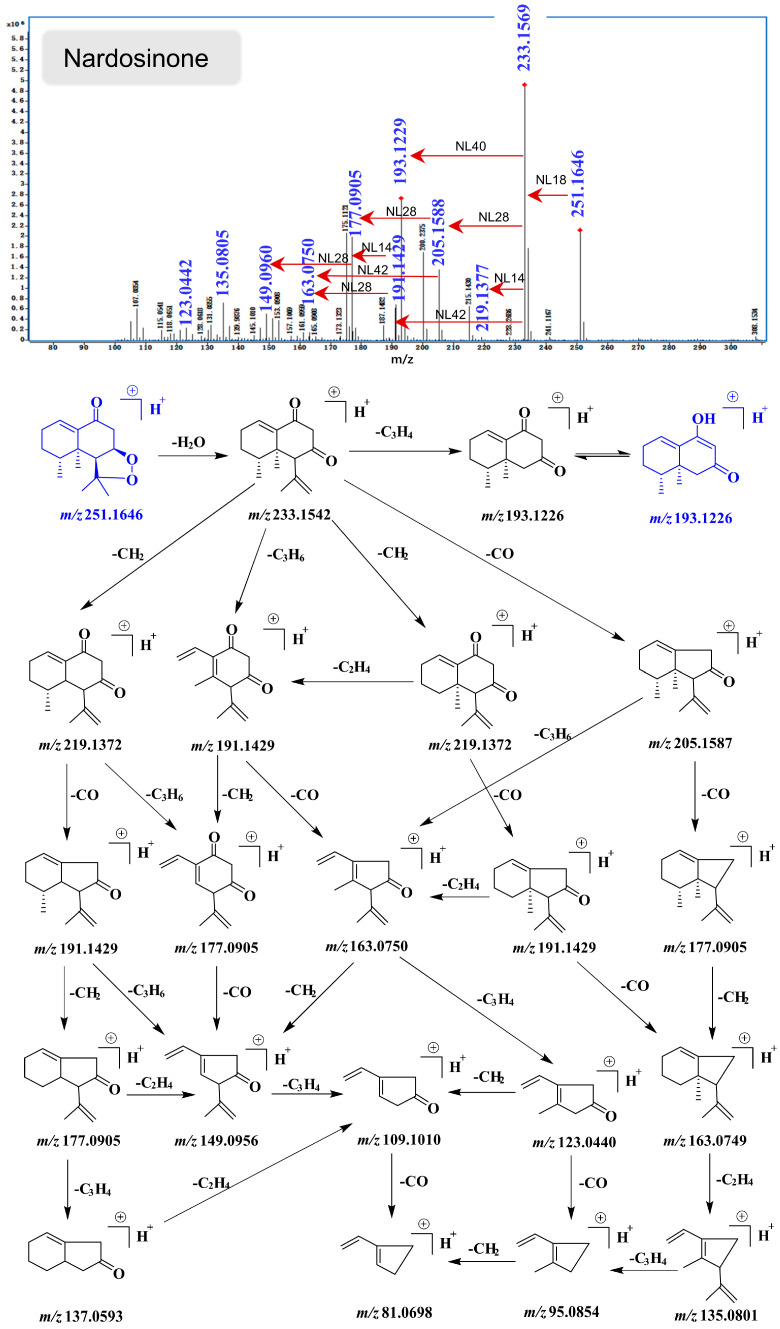
MS spectra and the proposed (+)–ESI–MS spectrometric fragmentation pattern of nardosinone.

**Figure 16 molecules-28-05382-f016:**
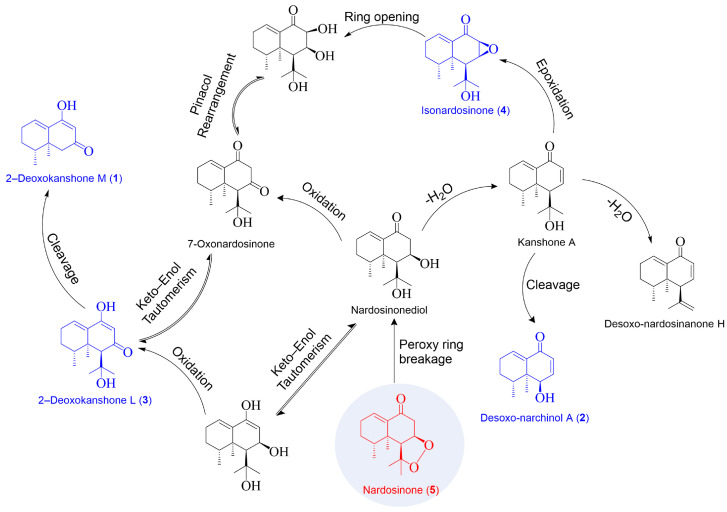
The plausible degradation pathway of nardosinone.

**Table 1 molecules-28-05382-t001:** NMR data for 2–deoxokanshone M in DMSO-*d*_6_.

Position	*δ*_H_/ppm (*J* in Hz) *^a^*	*δ*_C_, Type *^b^*	HMBC	^1^H–^1^H COSY
1	6.49 (1H, t, 4.0)	130.6, CH	C–3, 5, 9, 10	H–2
2	2.20 (2H, m)	25.6, CH_2_	C–1, 3, 4, 9, 10	H–1, H–3
3	1.46 (2H, m)	25.3, CH_2_	C–1, 2, 5, 11	H–2, H–4
4	1.62 (1H, m)	38.5, CH	C–3, 5, 6, 11, 12	H–3, H–11
5	–	37.4, C	–	–
6	2.28 (2H, m)	49.2, CH2	C–5, 7, 8, 10, 12	–
7	–	196.3, C	–	–
8	5.23 (1H, s)	102.0, CH	C–6, 7, 9, 10	–
9	–	167.7, C	–	–
10	–	136.1, C	–	–
11	0.87 (3H, d, 6.8)	15.4, CH_3_	C–3, 5	H–4
12	0.92 (3H, s)	19.4, CH_3_	C–4, 6, 10	–
-OH	10.83 (1H, s)	–	–	–

*^a^* Measured at 600 MHz. *^b^* Measured at 150 MHz.

**Table 2 molecules-28-05382-t002:** NMR data for 2–deoxokanshone L in DMSO-*d*_6_.

Position	*δ*_H_ (*J* in Hz) *^a^*	*δ*_C_, Type *^b^*	HMBC	^1^H–^1^H COSY
1	6.53 (1H, br s)	130.4, CH	C–3	H–2
2	2.20 (2H, m)	25.5, CH_2_	C–3, 4	H–1, H–3
3	1.45 (2H, m)	25.8, CH_2_	C–1, 2, 4, 5, 14	H–2, H–4
4	2.68 (1H, m)	32.1, CH	C–2, 5, 15	H–3, H–14
5	–	40.1, C	–	–
6	2.30 (1H, s)	65.5, CH	C–5, 8, 10, 11, 13, 15	–
7	–	199.0, C	–	–
8	5.16 (1H, s)	101.7, CH	C–6, 9, 10	–
9	–	167.0, C	–	–
10	–	136.8, C	–	–
11	–	71.7, C	–	–
12	0.98 (3H, s)	32.9, CH_3_	C–6, 11, 13	–
13	1.10 (3H, s)	25.2, CH_3_	C–6, 11, 12	–
14	0.89 (3H, d, 7.8)	16.7, CH_3_	C–3, 4, 5	H–4
15	0.90 (3H, s)	22.9, CH_3_	C–4, 5, 6, 10	–

*^a^* Measured at 600 MHz. *^b^* Measured at 150 MHz.

**Table 3 molecules-28-05382-t003:** Parameters of calibration curves for the simultaneous quantification of the five marker compounds.

Compound	Regression Equation	*r* ^2^	Linear Range (μg/mL)	LOD (μg/mL)	LOQ (μg/mL)
Desoxo-narchinol A	*y* = 7310.7*x* + 2169.9	0.9999	0.9766–62.5	0.216	0.720
2–Deoxokanshone L	*y* = 5960.1*x* + 1819.5	1	0.9766–250	0.208	0.693
Isonardosinone	*y* = 12,564*x* – 20,490	0.9996	1.9531–250	0.403	1.342
2–Deoxokanshone M	*y* = 20,365*x* – 47,345	0.9999	0.9766–1000	0.264	0.880
Nardosinone	*y* = 5926.7*x* + 20,602	0.9999	0.9766–1000	0.244	0.813

## Data Availability

The data are contained within the article and Appendix A.

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
