# Peer review of "Degradation Profiling of Nardosinone at High Temperature and in Simulated Gastric and Intestinal Fluids"

_molecules, 2023, doi:10.3390/molecules28145382_

Round 1

Reviewer 1 Report

The present study "Degradation Profiling of Nardosinone in High Temperature, and Simulated Gastric and Intestinal Fluids " focused on the degradation of nardosinone in refluxing boiling water and simulated gastric and intestinal fluids. Chemical investigation of the refluxing products of nardosinone resulted in the isolation and identification of four significant compounds, namely 2-deoxokanshone M (64.23%), desoxo-narchinol A (2, 2.17%), 7-oxonardosinenol (3, 1.10%), and isonardosinone (4, 3.44%). 2-Deoxokanshone M was determined to be a novel C 12 norsesquiterpenoid featuring an α, β-unsaturated enol group in its structure. It was found to possess significant vasodilatory activity without any anti-neuroinflammatory activity. Subsequently, UHPLC-PDA and UHPLC-DAD/Q-TOF MS were integrated for profiling the degradation products of nardosinone in high temperature (HT), simulated gastric fluid (SGF), simulated intestinal fluid (SIF), SGF-A, and SGF-B. The results suggested that a greater extent of degradation of nardosinone occurred under elevated temperature and simulated gastric fluid. The transformation pathway of 2-deoxokanshone M was deduced rationally, and the MS fragmentation patterns of nardosinone and the plausible degradation pathway were suggested accordingly. This study provided scientific evidence for the quality control and rational usage of nardosinone-related products, which are of great value in exploiting the therapeutic potential of nardosinone.
This work's findings provide insights into the degradation of nardosinone and can help guide the quality control of nardostachys jatamansi-related products.
Overall, this is a critical study that provides a valuable future direction to explore the stability and therapeutic potential of nardosinone. However, Suggested changes should be made to incorporate the implications and significance of this work properly.
-To further explore the degradation pattern and chemistry of nardosinone and its potential applications, it is suggested to add more detailed descriptions for perspective to conduct further studies to characterize the components of N. jatamansi and to investigate the mechanisms and dynamics of degradation pathways upon exposure to different environmental factors.
-Additionally, a comprehensive evaluation of the efficacy and safety of nardosinone, should be performed to verify further its potential therapeutic efficacy in treating various diseases.
-It is beneficial to include a brief description of the NMR measurements in the Materials and Methods section, which should include details such as pulse width, scan number, spectral width, relaxation delay, etc.
-In the introduction, it is necessary to discuss recent work on biologically active terpenoids [ 10.3389/fphar.2018.00116,   10.1021/acsabm.1c00550, 10.1016/j.molliq.2019.112366] to emphasize the study's relevance.
-On the labels of the NMR spectra in Figures 3, 10, 11, the units of measurement of chemical shifts (ppm) must be indicated.
-Finally, the further description should be done on the possible synergistic or antagonistic effects of components in the N. jatamansi extract and the products thereof.

Author Response

  1. To further explore the degradation pattern and chemistry of nardosinone and its potential applications, it is suggested to add more detailed descriptions for perspective to conduct further studies to characterize the components of N. jatamansi and to investigate the mechanisms and dynamics of degradation pathways upon exposure to different environmental factors.

 Reply: Thanks for your suggestion. We have added relevant descriptions at the end of this manuscript. On the one hand, in this study, we have isolated and identified the degradation products of nardosinone, and on this basis, we've also speculated on the degradation pattern of nardosinone upon exposure to different environmental factors. On the other hand, we have made reasonable speculations on the fragmentation pattern of nardosinane-type sesquiterpenoids by UPLC-DAD/Q-TOF MS. This can greatly improve the accuracy of the study on nardosinone. For example, Jing Zhang et al. speculated on the metabolites and cleavage pathways of nardosinone (Zhang et al., 2022), but the presumed degradation products differed slightly from those isolated and corroborated in this study. In addition, nardosinone, one of the major constituents of N. jatamansi, is a typical nardosinane-type sesquiterpenoid. The study of the degradation products and patterns of nardosinone will facilitate the identification of the components of N. jatamansi extracts or fractions, and the elucidation of dynamic mechanisms. Actually, with the aim of seeking the perspective values of our study as reported in this manuscript, we are trying to establish a specific and fast UHPLC-MS/MS method (by scanning CCS values for differentiating the isomers) for further studies on how the degradation pattern of nardosinone affect the investigation of the mechanisms and dynamics of degradation pathways on N. jatamansi upon exposure to different environmental factors.

 Related references:

Zhang, J., Lv, Y., Zhang, J., Bai, Y.S., Li, M.Y., Wang, S.Q., Wang, L.L., Liu, G.X., Xu, F., Shang, M.Y., Cai, S.Q., 2022. Analysis of In Vivo Existence Forms of Nardosinone in Mice by UHPLC-Q-TOF-MS Technique. Molecules. 27, 7267. https://doi.org/10.3390/molecules27217267.

  1. Additionally, a comprehensive evaluation of the efficacy and safety of nardosinone, should be performed to verify further its potential therapeutic efficacy in treating various diseases.

 Reply: Thanks. It is definitely a great suggestion. Previous studies have reported that nardosinone

possesses various activities, such as anti-inflammatory effects, antidepressant, cardioprotective, anti-neuroinflammatory, vasodilatory, anti-arrhythmic, anti-periodontitis, anti-hypertrophic effect in cardiomyocytes, antibacterial and insecticidal effects, enhances the activity of the nerve growth factor and promotes neural stem cells to proliferate and differentiate (Wen et al., 2021). As for the safety of nardosinone, however, to date, there are few studies on its toxicity instead of that of the originated plant of N. jatamansi. As is reported, the extracts and the major constituents isolated from N. jatamansi have been often concluded to show either low or no toxicities in acute toxicity tests, revealing the safety property of this herb (Wang et al., 2021). Hence, we are about to investigate the neurotoxicity potential of nardosinone to verify its safety. In addition, the stability and degradation products of nardosinone were investigated in this work, it can be regarded as a prerequisite for examining its safety.

Related references:

Wang, M., Yang, T.T., Rao, Y., Wang, Z.M., Dong, X., Zhang, L.H., Han, L., Zhang, Y., Wang, T., Zhu, Y., Gao, X.M., Li, T.X., Wang, H.Y., Xu, Y.T., Wu, H.H., 2021. A review on traditional uses, phytochemistry, pharmacology, toxicology and the analytical methods of the genus Nardostachys. J Ethnopharmacol. 280, 114446. https://doi.org/10.1016/j.jep.2021.114446.

Wen, J., Liu, L., Li, J., He, Y., 2021. A review of nardosinone for pharmacological activities. Eur J Pharmacol. 908, 174343. https://doi.org/10.1016/j.ejphar.2021.174343.

  1. It is beneficial to include a brief description of the NMR measurements in the Materials and Methods section, which should include details such as pulse width, scan number, spectral width, relaxation delay, etc.

 Reply: Thanks. We have added detailed parameters of the NMR measurements in Supplementary Table S1, including spectrometer frequency (SF), acquisition time (AQ), relaxation delay (RD), pulse width (P1), spectral width (SWH), line broadening (LB), Fourier transform (FT), scan number (NS), etc.

  1. In the introduction, it is necessary to discuss recent work on biologically active terpenoids [10.3389/fphar.2018.00116, 10.1021/acsabm.1c00550, 10.1016/j.molliq.2019.112366] to emphasize the study's relevance.

 Reply: Thanks a lot. Since our work focused on the stability of nardosinone, a sesquiterpenoid from N. jatamansi featuring a unique peroxide bridge. So, we think it is necessary to discuss the research findings on nardosinone instead of those of the biologically active terpenoids. We appreciate that you've provided three excellent publications, however, all of them referred to recent research on monoterpenoids, which we think is not reasonably relevant to our work. We'd like to say sorry for this, and we hope it doesn't affect the positive impression of our manuscript, thanks.

  1. On the labels of the NMR spectra in Figures 3, 10, 11, the units of measurement of chemical shifts (ppm) must be indicated.

 Reply: Thanks for your reminder. In this edition of our manuscript, we have added the units of measurement of chemical shifts (ppm) for the NMR spectra in Figures 3, 10, 11.

  1. Finally, the further description should be done on the possible synergistic or antagonistic effects of components in the N. jatamansi extract and the products thereof.

 Reply: Thank you for your suggestion. There is no doubt that it is beneficial for better medicinal application to clear the synergistic or antagonistic effects of components in the N. jatamansi extract. Unfortunately, as far as the current research data we can reach, our ongoing experiments haven't given us any confidence to make any accurate speculations on the possible synergistic or antagonistic effects of components in the N. jatamansi extract and the products thereof. At a later stage, we are designing a detailed and relevant experiment to investigate this, and it can be expected that the possible synergistic or antagonistic effects of components will be unveiled soon.

Reviewer 2 Report

The authors reported a full degradation studies of Nardosinone in High Temperature, and Simulated Gastric and Intestinal Fluids.  However, many changes are recommended.

1-     In the abstract more details about types multiple diseases that Nardosinone could be used for.

2-     Full names for abbreviations [HT, SGF, SIF, IEFPCM, ECD] should be stated when first mentioning. Besides, abbreviation list is recommended.

3-     In introduction , the following articles should be cited

A.     https://www.sciencedirect.com/science/article/pii/S0014299921004969

A review of nardosinone for pharmacological activities, Author links open overlay panelJiawei Wen, Linqiu Liu, Junjun Li, Yang He

B.     https://www.ncbi.nlm.nih.gov/pmc/articles/PMC9653913/

Jing Zhang,1 Yang Lv,1 Jing Zhang,1 Yu-Sha Bai,1 Meng-Yuan Li,2 Shun-Qi Wang,1 Li-Li Wang,2 Guang-Xue Liu,1 Feng Xu,1,* Ming-Ying Shang,1 and Shao-Qing Cai1,*

Analysis of In Vivo Existence Forms of Nardosinone in Mice by UHPLC-Q-TOF-MS Technique

4-     The position of table 1 should be shifted after line 187.

5-     NMR should be described as 1H-NMR in all mentioning.

6-     In table 2,  LOQ values should be equal or less than the minimum value of linearity range. This is very important point for analysts.

7-     future plan and study limitation should be highlighted

Best wishes

Author Response

Reviewer #2: The authors reported a full degradation studies of Nardosinone in High Temperature, and Simulated Gastric and Intestinal Fluids. However, many changes are recommended.

  1. In the abstract more details about types multiple diseases that Nardosinone could be used for. 

Reply: Thank you very much for this suggestion. In this edition of our manuscript, the sentence of "Nardosinone, a predominant bioactive product from Nardostachys jatamansi DC, is well-known for its promising therapeutic potentials, such as anti-inflammatory, antidepressant, cardioprotective, anti-neuroinflammatory, anti-arrhythmic, anti-periodontitis, etc." is added to address the multiple types of diseases that nardosinone could be used for.

  1. Full names for abbreviations [HT, SGF, SIF, IEFPCM, ECD] should be stated when first mentioning. Besides, abbreviation list is recommended. 

Reply: Thanks for your kind reminder. We have stated the full name for all abbreviations when they were first mentioned. And an abbreviation list has been added accordingly.

  1. In introduction, the following articles should be cited: 1) https://www.sciencedirect.com/science/article/pii/S0014299921004969,A review of nardosinone for pharmacological activities, Author links open overlay panelJiawei Wen, Linqiu Liu, Junjun Li, Yang He; 2) https://www.ncbi.nlm.nih.gov/pmc/articles/PMC9653913/,Jing Zhang,1 Yang Lv,1 Jing Zhang,1 Yu-Sha Bai,1 Meng-Yuan Li,2 Shun-Qi Wang,1 Li-Li Wang,2 Guang-Xue Liu,1 Feng Xu,1, Ming-Ying Shang,1 and Shao-Qing Cai1, Analysis of In Vivo Existence Forms of Nardosinone in Mice by UHPLC-Q-TOF-MS Technique 

Reply: We sincerely appreciate the valuable comments. We have cited the two articles you've kindly provided in this edition of our manuscript.

  1. The position of table 1 should be shifted after line 187. 

Reply: Thank you for your suggestion, and we've shifted Table 1 to the right place.

  1. NMR should be described as 1H-NMR in all mentioning. 

Reply: Thank you very much for this reminder. As far as we know, NMR includes not only 1D NMR (1H NMR, 13C NMR), but also 2D NMR (HSQC, HMBC, 1H-1H COSY). Thus, if NMR was not described or referred in particular as 1H NMR or any other types of NMR experiments, it then represented the extensive NMR experiments including 1D NMR and/or 2D NMR. We've carefully corrected it throughout the manuscript as well as the supplementary materials, thanks.

  1. In table 2, LOQ values should be equal or less than the minimum value of linearity range. This is very important point for analysts. 

Reply: We sincerely appreciate your suggestion. We have reset the linearity range to afford a new regression equation for a more accurate data analysis. Thanks again.

  1. Future plan and study limitation should be highlighted 

Reply: Thanks for your valuable suggestion. As you can see, we have presented our future plan and limitation of this study in the conclusion as follows:

"Our study preliminarily sheds light on the stability of nardosinone. Future investigations are still needed for the analysis on the metabolites of nardosinone and the components absorbed in the blood. This study may contribute to providing reasonable guidance for preparation production and quality control of nardosinone, N. jatamansi, and the prepared patent Chinese medicines containing N. jatamansi."

Round 2

Reviewer 2 Report

the authors did all the required recommendations. i appreciate their professionalism. 

the paper could be published in the current form